# CCHamide-2 Signaling Regulates Food Intake and Metabolism in *Gryllus bimaculatus*

**DOI:** 10.3390/insects13040324

**Published:** 2022-03-25

**Authors:** Zhen Zhu, Maho Tsuchimoto, Shinji Nagata

**Affiliations:** Department of Integrated Biosciences, Graduate School of Frontier Sciences, The University of Tokyo, Chiba 277-8562, Japan; 9961445003@edu.k.u-tokyo.ac.jp (Z.Z.); nesslucas@icloud.com (M.T.)

**Keywords:** CCHamide-2 signaling, feeding, metabolism, *Gryllus bimaculatus*

## Abstract

**Simple Summary:**

CCHamide-2 is a newly identified neuropeptide associated with the regulation of feeding behavior as a brain-gut peptide in insects. Data revealed CCHamide-2 signaling inhibited feeding along with increasing circulating carbohydrate and lipid levels in the two-spotted crickets, *Gryllus bimaculatus*. Consequently, we demonstrated that the signaling involving CCHamide-2 and its receptor contributes to the regulation of feeding and metabolism in the neuropeptide-mediated regulatory network of energy homeostasis in insects.

**Abstract:**

Neuropeptides play vital roles in energy homeostasis in insects and other animals. Although the importance of the regulatory network of neuropeptides in feeding and metabolism has been illuminated, a complete understanding of the mechanisms has not been addressed as many factors are involved in the regulation. CCHamide-2 is a newly identified brain-gut neuropeptide that regulates feeding behavior in several insect species including *Drosophila melanogaster*. However, little is known about the mechanisms controlling the feeding-related behavior and metabolic functions modulated by CCHamide-2 in other insects. In this study, we addressed the functions of CCHamide-2 in the two-spotted cricket, *Gryllus bimaculatus*, which was used as the experimental material to research the mechanisms of feeding and metabolism in this omnivorous insect species. Knockdown crickets by RNA interference against *GbCCHamide-2R* increased the amount of food intake, while injection of chemically synthetic GbCCHamide-2 peptide reduced the amount of food intake. Further, knockdown and peptide injection experiments revealed that GbCCHamide-2 signaling increased the concentrations of circulating lipids and carbohydrates, and the carbohydrate-rich diet increased the transcript levels of *GbCCHa-2R.* Moreover, GbCCHa-2 injection decreased the transcript level of *Gbilp*. By contrast, GbCCHamide-2 signaling did not affect nymphal growth or the transcript level of *GbAKH*, as well as feeding preferences. Taken together, CCHamide-2 signaling in *G. bimaculatus* regulates food intake associated with alterations in lipid and carbohydrate levels in hemolymph.

## 1. Introduction

Feeding and metabolism are well-regulated in organisms for maintenance of energy homeostasis, which is essential for their survival and health [1]. Dysregulation of feeding and metabolism triggers physiological distress and is associated with the development of a series of metabolic diseases [2,3]. Neuropeptides, which are the largest group of key signaling molecules in multicellular organisms including insect species, greatly contribute to the regulation of feeding and metabolism [4,5].

To date, several neuropeptides have been characterized as modulators of feeding and metabolism in insects. For example, insulin-like peptides (ILPs) are recognized as crucial signaling factors in the regulation of energy homeostasis by modulating circulating carbohydrate and lipid levels in invertebrates as well as vertebrates [6,7,8]. Similarly, neuropeptide F is a functional and structural analog of neuropeptide Y, and adipokinetic hormone (AKH) is a functional analog of glucagon, which contribute significantly to energy homeostasis [9,10,11,12]. Although increasing numbers of the identified neuropeptides in insect species have been found, the mechanisms underlying the neuropeptides-mediated regulatory network of feeding and metabolism have not been completely elucidated.

CCHamide has first been reported in 2000 as an arthropod-specific enigmatic neuropeptide associated with the regulation of feeding [13]. The mature peptide of CCHamide contains a C-terminal histidine-amide residue and two cysteine residues forming an intramolecular disulfide bond. Most arthropods contain only one CCHamide subtype within one species; however, insects always contain two subtypes of CCHamide (CCHa-1 and CCHa-2) probably generated via the duplication of CCHamide that is present in the early-diverging lineages of insect [14,15]. These two CCHamides, CCHa-1 and CCHa-2 specifically activate their corresponding receptors belonging to the classes of G protein-coupled receptors (GPCRs) having characteristic seven transmembrane regions [15,16], which have been identified in *Drosophila melanogaster* and *Bombyx mori* [15,17]. Indeed, CCHa-1 and -2 signaling function via different mechanisms by the functional analyses predominantly performed in *D. melanogaster*. DmCCHa-1 is involved in the regulation of olfactory sensitivity [18], circadian activity [19], and arousal responses via protein-rich diet [20]. In contrast to the function of DmCCHa-1, DmCCHa-2 is believed to function as an orexigenic peptide [21,22]. Therefore, CCHa-1 and CCHa-2 are thought to be the different bioactive peptides even though they have the structural similarities as these peptides are designated.

In *D. melanogaster*, *CCHa-2* mutant flies display reduced food consumption, while the rescued mutants exhibit normal feeding activity. Additionally, CCHa-2 signaling regulates growth by activating insulin-producing cells in the brain [23]. The effects of nutrients have also been investigated by refeeding a yeast-based diet or glucose after starvation, resulting in an increased mRNA level of *DmCCHa-2* in the third-instar *D. melanogaster* larvae [23]. Meanwhile, the diet with lower protein concentration reduces transcript level of *CCHa-2* in *D*. *melanogaster* larvae [24], seemingly contrasted to the effect of feeding yeast. The expression of CCHa-2 may be modulated by the dietary nutrients and their concentrations, leading to the complicated control of CCHa-2. Therefore, more efforts should be taken to elucidate the regulatory relationship between CCHa-2 signaling and the metabolism of these nutrients.

In insects other than *D. melanogaster*, the functions of CCHa-2 signaling have been reported in two species so far; CCHa-2 is associated with the regulation of post-prandial diuresis and inhibits serotonin-stimulated transcellular Na^+^ transport across the anterior midgut of the kissing bug, *Rhodnius prolixus* [25,26]; *CCHa-2R* knockdown reduces food intake and reproduction of the pea aphid, *Acyrthosiphon pisum* [27]. To illuminate the general roles of CCHamide in insects, we used the orthopteran species, the two-spotted cricket, *Gryllus bimaculatus*, as an experimental material in this study. This species is considered to be an optimal model for exploring the regulatory mechanisms underlying feeding and metabolism, since the relatively large body size facilitates observation of feeding and analyses by surgical approaches [28]. RNA interference (RNAi) was used as it presents an efficient technique to perform loss-of-function analyses of the genes of interest [29].

In the current study, *GbCCHa-2R* knockdown by RNAi and treatment with GbCCHa-2 peptide were performed to clarify the modulatory roles of GbCCHa-2 signaling in feeding behavior as well as lipid and carbohydrate metabolism. Next, we explored the possibility whether GbCCHa-2 signaling would contribute to the selective behavior for the concentrations and types of three primary macronutrients in diets.

## 2. Materials and Methods

### 2.1. Insects

*G. bimaculatus* were reared in plastic containers (55 × 39 × 32 cm) at 27 ± 1 °C under a photoperiod of 16 h light and 8 h darkness. The diets were composed of rabbit food ORC4 (Orient Yeast, Tokyo, Japan) and cat food (Purina^®^, Friskies^®^) at a weight ratio of 4:1. Eighth-instar male nymphs (final instar nymphs) were used for the following experiments.

### 2.2. Tissue-Distribution Analyses Using Reverse Transcription-PCR (RT-PCR)

Tissues including brain, corpora allata, corpora cardiaca, subesophageal ganglion, thoracic ganglia, abdominal ganglia, terminal abdominal ganglion, foregut, midgut, hindgut, fat body, muscle, testis, and Malpighian tubules were dissected from eighth-instar male nymphs on the second day after molting. Total RNA was extracted using TRI reagent^®^ (Molecular Research Center, Inc., Cincinnati, OH, USA) according to the manufacturer’s protocol. Contaminant genomic DNA was removed from total RNA by mixing the sample with RQ DNase I (Promega Co., Madison, WI, USA). The RNA was further purified by phenol/chloroform extraction and ethanol precipitation, and finally dissolved in 0.1% diethylpyrocarbonate (DEPC)-treated water to a concentration of 100 ng/µL. The resulting RNA was used as the template for no-RT reactions to ensure no DNA contamination, or reverse-transcribed using ReverTra Ace^®^ (Toyobo Co. Ltd., Osaka, Japan) to obtain cDNA, which was used as the template DNA to amplify the partial fragments of *Gb**CCHa-2* and -*2R*, as well as the internal control gene, *elongation factor* (*EF*) (GenBank accession number: ABG01881.1). The specific primers were listed in Table 1 for *GbCCHa-2* and *-2R* based on the putative sequences obtained from the in-house transcriptional data based on the previous studies [28,30]. The obtained sequences of *GbCCHa-2* and *-2R* were identical with Transcriptome Shotgun Assembly sequences, GFMG02176256 and GFMG02051271, in GenBank, respectively. Fragments of *EF* and *Gb**CCHa-2R* were amplified using Go Taq Green Master Mix (Promega). The *Gb**CCHa-2* fragment was amplified using the LA Taq Kit (Takara Bio, Shiga, Japan) owing to the high GC contents in its sequence. The following polymerase chain reaction (PCR) program was used for amplifying all samples: initial denaturation at 94 °C for 2.5 min; 35 cycles of 30 s at 94 °C, 30 s at 55 °C, and 30 s at 72 °C, and a final step at 72 °C for 10 min. The PCR products were purified using Wizard SV Gel and PCR Clean-up System (Promega), and the purified PCR products were inserted into pGEM-T vector (Promega). The nucleotide sequences of the inserted cDNA in the plasmid were confirmed using 3500 Genetic Analyzer (Applied Biosystems, Waltham, MA, USA).

### 2.3. RNA Interference

A knockdown-target fragment DNA of *GbCCHa-2R* was amplified by PCR using the plasmid with an insert fragment of *GbCCHa-2R* with primers containing the T7 promoter sequences (Table 1). The PCR product flanking with T7 sequences at both 5′- and 3′- ends were purified to synthesize double-stranded RNA (dsRNA) by using T7 polymerase from both 5′- and 3′- ends (TaKaRa Bio). The synthesized dsRNA was treated with RQ DNase I (TaKaRa Bio) to remove template DNA from the reaction mixture, then purified by phenol/chloroform extraction and ethanol precipitation, and finally dissolved in DEPC-treated RNase-free water to a concentration of 2.5 µg/µL. Next, the dsRNA was denatured for 5 min at 100 °C and cooled to the room temperature for annealing. Ten micrograms (4 µL) of ds*GbCCHa-2R* were injected into the abdomen of newly emerged eighth-instar male nymphs. *DsRed2* was used as a control gene in this study. Total RNAs from the central nervous system (CNS) including brain, subesophageal ganglion, thoracic ganglia, abdominal ganglia and terminal abdominal ganglion, and gut were extracted on the third day after injection and were reverse transcribed to cDNAs, which were used to evaluate RNAi efficiency by quantitative reverse transcription PCR (qRT-PCR). qRT-PCR was performed as the following description. The mixture of cDNA, SYBR Premix Ex Taq II (Tli RNaseH Plus; TaKaRa, Shiga, Japan) and specific primers (Table 1) was reacted on a Thermal Cycler Dice Real Time System TP850 (TaKaRa, Shiga, Japan) with the program: 30 s at 95 °C, 40 cycles of 5 s at 95 °C, and 30 s at 55 °C; ended by a dissociation curve analysis at 95 °C for 15 s, at 55 °C for 30 s, and at 95 °C for 15 s. *G. bimaculatus ß-actin* (GenBank accession number: AB626808.1) was used as a reference gene. The relative transcript levels were calculated using 2^–ΔΔCT^ method.

### 2.4. Chemical Synthesis of GbCCHa-2

The synthesis of peptide GbCCHa-2 (GCSAFGHSCFGGH-NH_2_) was carried out according to the fluorenyl methoxycarbonyl (Fmoc) method by using Rink amide resin (Merck Millipore, Darmstadt, Germany) [30,31]. A mixture of Fmoc amino acids (Kokusan Chemical, Tokyo, Japan); 1-hydroxybenzotriazole, and *N,N*’-dicyclohexylcarbodiimide was allowed to react in *N*- methylpyrrolidone for successive elongation of the polypeptide chain attached to the resin. The Fmoc protecting group was removed by shaking the mixture together with 20% piperidine in dimethylformamide. The assembled peptide was decoupled with the resin and protecting groups via treatment with a mixture of trifluoroacetic acid (TFA); dichloromethane; anisole; trimethylsilyl bromide, and 3,4-ethoxylene dioxythiophene at a ratio of 10:5:2:2:1 (*v*/*v*/*v*/*v*/*v*) at 4 °C overnight. The crude linear peptide was mixed with diethyl ether, followed by centrifugation to precipitate the peptide which was then dissolved in 0.1% TFA and purified using a reversed-phase Sep-Pak C18 cartridge. The column was washed with 90% acetonitrile aqueous solution containing 0.1% TFA. After charging the cartridge with 0.1% TFA aqueous solution, the sample solution was flowed through the column to adsorb the peptide. Sixty percent acetonitrile in 0.1% TFA aqueous solution was used to elute the synthetic peptide. The eluate was purified using a reversed-phase high-performance liquid chromatography (HPLC) (Jasco SC-802, PU-880, UV-875; Jasco Int., Tokyo, Japan) on a Senshu Pak PEGASIL-300 ODS column (4.6 mm i.d. × 250 mm; Senshu Kagaku, Tokyo, Japan) with a linear gradient of 0–60% acetonitrile containing 0.1% TFA over 40 min. The purified peptide was confirmed by matrix-assisted laser desorption ionization time-of-flight mass spectrometry (MALDI-TOF MS). The intramolecular disulfide bond of GbCCHa-2 peptide was artificially formed by dissolving the lyophilized peptide in 100 mM Tris-HCl (pH 8.0), followed by air-oxidation with shaking. The oxidized peptide was purified using the reversed-phase HPLC under the same condition as described above. The amount of synthetic GbCCHa-2 was calculated based on the peak area of bovine serum albumin (BSA) by the absorption at 225 nm in the reversed-phase HPLC analyses. The prepared peptide was used after being dissolved in water for the following assays. In the peptide-injection experiments but not in the dose-response assay for food intake, 1 µg (10 µL) of peptides were injected. Ten microliters of water was injected as the control.

### 2.5. Food Intake Assays

Crickets were isolated in plastic containers with freely available excess food and water under the same conditions as the rearing conditions. The number of excreted fecal pellets represented the amount of food consumed. In addition, crickets used in the experiments were weighed at the beginning and the end of the third day after dsRNA treatment, and 12 h after peptide administration.

### 2.6. Measurement of Lipid and Carbohydrate Concentrations

Lipids and carbohydrates (trehalose and glucose) from the hemolymph, as well as lipids and glycogen from fat body were extracted as described previously [32]. At the end of the third day after dsRNA injection and at the end of 12 h after peptide administration, fat body, and 5 µL of hemolymph were collected from the abdomen of the crickets. Fat body was collected in 50 µL water, lyophilized, and then weighed.

The sulfo-phospho-vanillin method was used for the measurement of total lipid content [30]. Cholesterol (Sigma-Aldrich Japan, Tokyo, Japan) in the same solvent at the concentrations of 100, 20, 4, 0.8, and 0.16 µg/µL was used for generating the standard curve.

The hexokinase/glucose-6-phosphate dehydrogenase method was used to measure the concentration of free carbohydrates [30]. Carbohydrate concentration was measured after the enzymatic conversion from trehalose into glucose by trehalase (Sigma-Aldrich). The standard curve was generated using 100, 10, 1, 0.1, and 0.01 µg/µL glucose in water. The final data indicated the concentrations of trehalose plus glucose.

Glycogen concentration was measured using anthrone method as previously described [32]. Glycogen (Boehringer Mannheim GmbH, Germany) solutions at the concentrations of 100, 20, 4, 0.8, and 0.16 µg/µL were used for a standard.

### 2.7. Food Choice Assay

For the two-choice assay, crickets were fed on a general rearing diet for two days after injection of dsRNA. On the third day, individual crickets were provided with one of the following paired diets for 24 h: high-lipid (casein 2.3 g, soybean oil 8.6 g, and dextrin 2.3 g per 100 g diet) and low-lipid diets (soybean oil 1.1 g instead of 8.6 g); high-protein (casein 19 g, soybean oil 1.1 g, and dextrin 2.3 g per 100 g diet) and low-protein diets (casein 2.3 g instead of 19 g); high-carbohydrate (casein 2.3 g, soybean oil 1.1 g, and dextrin 19 g per 100 g diet) and low-carbohydrate diets (dextrin 2.3 g instead of 19 g). Cellulose powder (CLEA Japan Inc., Japan) was added to compensate for the weight of the experimental diets. The prepared experimental diets were provided after peptide application, and the nymphs were allowed to feed for 12 h. The weight of each food item before and after feeding was measured. Food preference was evaluated as the preference index (PI) by the ratio of each diet’s consumption to the examined diets.

For the three-choice assay, there were three different diets containing only lipid, or carbohydrate, or protein as the nutrient in each diet, respectively. We then placed these three different diets in front of the examined crickets. All diets prepared contained the same caloric content of 4 kJ/g. The proportion of each consumed diet was then calculated.

## 3. Results

### 3.1. Tissue Distributions of GbCCHa-2 and GbCCHa-2R

The spatial distributions of *GbCCHa-2* and *-2R* expression were analyzed. RT-PCR analyses revealed that *GbCCHa-2* transcript was predominantly detected in the CNS, midgut, and testis (Figure 1). By contrast, *GbCCHa-2R* transcript was widely distributed in most tissues based on the RT-PCR results, especially in the CNS and gut (Figure 1).

### 3.2. Effects of GbCCHa-2 Signaling on Food Intake and Body Weight

To elucidate the effects of GbCCHa-2 signaling on feeding behavior, we knocked down *GbCCHa-2R* in crickets by RNAi. dsRNA encoding GbCCHa-2R (ds*GbCCHa-2R*) was injected into newly emerged eighth-instar male nymphs. qRT-PCR analyses revealed that *GbCCHa-2R* transcript levels in the CNS and gut of ds*GbCCHa-2R*-injected nymphs were significantly decreased compared to those of the control group, with 42% and 59% reduction, respectively (Figure 2). Using these ds*GbCCHa-2R*-treated crickets, we first evaluated the effects of GbCCHa-2 signaling on food intake by counting the number of fecal pellets according to the previous report [28]. As a result, *GbCCHa-2R*-knockdown crickets excreted robustly larger number of feces, indicating an increase in food intake (Figure 3a). To examine the effects of GbCCHa-2 peptides on food intake, we next injected different doses of chemically synthesized GbCCHa-2 (10 ng, 100 ng, and 1000 ng) into cricket nymphs. At 12 h after peptide injection, numbers of fecal pellets were decreased compared to the control in a dose-dependent manner. The significant inhibitory effects on excretion were observed by injection of more than 100 ng of GbCCHa-2 (Figure 3a). In contrast to the alteration of food intake by GbCCHa-2 signaling manipulation, ds*GbCCHa-2R*-treated crickets did not show any difference in weight gain during the experimental period. However, weight gain of crickets was gradually decreased along with the increasing doses of the injected GbCCHa-2, yet no statistically significant difference was confirmed (Figure 3b). Collectively, these results indicate that GbCCHa-2 signaling might suppress food intake but might be dispensable for the control of body weight of crickets.

### 3.3. Effects of GbCCHa-2 Signaling on Lipid and Carbohydrate Concentrations in the Hemolymph and Fat Body

As the alteration in feeding is generally associated with a shift in the metabolic state, we measured the lipid and carbohydrate levels in the hemolymph and fat body of *GbCCHa-2R*-knockdown crickets. The concentration of hemolymph lipids in *dsGbCCHa-2R*-treated crickets was significantly lower than that in the control group. In contrast, GbCCHa-2 injection increased the lipid concentration in the hemolymph (Figure 4a). No alteration was observed in the carbohydrate level in the hemolymph by *GbCCHa-2R* knockdown compared to that in the control, whereas an increased level was observed in GbCCHa-2-applied crickets (Figure 4b). The levels of stored lipids and glycogen in fat body were not changed after either treatment (Figure 4c,d).

### 3.4. Effects of Dietary Nutrients on the Transcript Level of GbCCHa-2R

As described above, GbCCHa-2 signaling regulates the circulating lipid and carbohydrate concentrations. Then we investigated whether GbCCHa-2 signaling responds to the dietary nutrients. Four groups of crickets were fed on normal diet, lipid-, carbohydrate-, and protein-rich diets for 3 days. The transcript level of *GbCCHa-2R* in CNS was examined. The carbohydrate-rich diet significantly increased the transcript level of *GbCCHa-2R.* The lipid-rich diet caused an increased trend in the transcript level (Figure 5a). Since ILPs play a crucial role in the regulation of energy homeostasis, it is asked whether GbCCHa-2 signaling affects nutritional status via Gbilp signaling. On the third day after *GbCCHa-2R* RNAi or at the twelfth hour after GbCCHa-2 injection, the transcript level of *Gbilp* in CNS was examined. The results revealed that the transcript level of *Gbilp* was significantly increased in GbCCHa-2 peptide-applied cricket, while no alteration was observed in *GbCCHa-2R*-knockdown crickets (Figure 5b). The spatial distribution of *GbCCHa-2R* exhibited a weak signal in corpora cardiaca which is the only expressing location of *GbAKH* [30]. Nevertheless, manipulation of GbCCHa-2 signaling did not influence the transcript level of *GbAKH* (Figure 5c).

### 3.5. Effects of GbCCHa-2 Signaling on Food Preference

Insects select foods based on their nutritional status [33]. Given that GbCCHa-2 signaling appeared to be involved in the regulation of circulating lipids and carbohydrates (Figure 4), we investigated whether GbCCHa-2 signaling is directed to the nutritionally selective behavior along with the alteration of hemolymph lipid and carbohydrate levels. The two-choice assay was performed to test the nutritional preference of crickets. Displaying the diets containing different levels of macronutrients (lipids, carbohydrates, or proteins) revealed that the cricket nymphs preferred nutrient-rich diets (Figure 6a–c). Similarly, ds*GbCCHa-2R*-treated and GbCCHa-2-injected crickets did not change the trend of the preferences (Figure 6a–c). Furthermore, when those three kinds of diets composed of different macronutrients (lipids, carbohydrates, or proteins) were provided, all tested crickets preferred the lipid- and carbohydrate-rich diets compared to protein-rich diet, while the proportions of each consumed diet were not altered by *GbCCHa-2R* knockdown and peptide application (Figure 6d).

## 4. Discussion

In this study, we explored the function of CCHamide-2 in the crickets. According to the results of spatial distributions, *GbCCHa-2* was predominantly detected in the CNS and midgut (Figure 1), which is consistent with the previously demonstrated data in *D. melanogaster* and *R. prolixus* that CCHa-2 is expressed in the CNS and gut [25,34]. Therefore, the functions of CCHa-2 as a brain-gut neuropeptidyl factor seem to be conserved over insect species. Notably, *GbCCHa-2* transcript was also detected in the testis of cricket nymphs, which has not been reported in other insects. In the orange mud crab, *Scylla olivacea*, a transcript of *CCHa-2* is observed in the testis transcriptomic data [35], indicating that CCHa-2 may exert reproductive functions in arthropods. Although DmCCHa-2 has not been detected in the larval reproductive tissues, the dietary protein affects *DmCCHa-2* transcript level in male *D. melanogaster* larvae but has no effects in female larvae [24], illuminating gender-specific functions of DmCCHa-2. The expression of *GbCCHa-2* and *-2R* in the reproductive tissue, testis, implies that GbCCHa-2 signaling may also be associated with the phenotypic sexual dimorphism in crickets.

As observed in the present functional analyses that GbCCHa-2 signaling suppressed food intake but had no effect on weight gain (Figure 3), GbCCHa-2 may contribute slightly to the feeding behavior in crickets. Indeed, the inhibitory effect of GbCCHa-2 on food intake was exhibited by applying the higher doses of peptide (100 ng and 1000 ng). Moreover, GbCCHa-2 signaling increased the circulating lipid and carbohydrate concentrations, and the dietary carbohydrates stimulated the expression of *GbCCHa-2R* (Figure 4 and Figure 5a). However, GbCCHa-2 signaling did not change the nutritional preferences (Figure 6). In addition, GbCCHa-2 administration increased the transcript level of *Gbilp* (Figure 5b). There is one possibility to explain these findings; GbCCHa-2 signaling inhibits Gbilp signaling, which leads to the increased level of carbohydrates in hemolymph, eventually causing the suppression of feeding. By contrast, the effect of GbCCHa-2 on lipid metabolism were not involved in the transcriptional regulation of *AKH*, while the effect might be mediated by other regulatory factors (Figure 5c).

GbCCHa-2 signaling was found to be associated with the negative modulation of food consumption (Figure 3a). In contrast to the present study, CCHa-2 is characterized as an orexigenic peptide in *D. melanogaster* [21,22]. Additionally, *CCHa-2R* knockdown in the pea aphids exhibits decreased food intake [27]. The similar opposite effects are observed in the relationship between ilp and CCHa-2. The previous studies demonstrate *DmCCHa-2* mutants reduce mRNA levels of *Dilp* [21,23]*,* contrasted to the decreased mRNA level of *Gbilp* caused by GbCCHa-2 application. Therefore, the regulatory manner of food consumption via CCHa-2 signaling seems to be different for each species, although this signaling is commonly associated with feeding regulation.

In *D. melanogaster*, it has been reported that *DmCCHa-2* mutants show retarded development and decreased body size, whereas little influences in growth were observed in crickets treated with ds*GbCCHa-2R* or GbCCHa-2 peptide (Figure 3b). On the other hand, increased weight was observed only on the third day during the RNAi experiment and for the first 12 h during the peptide-treatment experiment; therefore, differences in growth rate cannot be reflected during such a short experimental period. In addition, as development and growth may be highly plastic, the growth abnormality caused by excessive or insufficient food consumption might be rescued by adjusting the efficiency of uptake of nutrients from the consumed and ingested food [36,37].

GbCCHa-2 injection increased the levels of lipids and carbohydrates in the hemolymph. In contrast, *GbCCHa-2R* knockdown reduced only the levels of circulating lipids (Figure 4). Additionally, the altered transcript level of *Gbilp* was observed in GbCCHa-2 peptide-applied crickets but not in *GbCCHa-2R*-knockdown crickets (Figure 5b). The unchanged carbohydrate concentrations in the hemolymph and expression of *Gbilp* via RNAi might be due to incomplete interference (Figure 2), or complementation by unidentified pathways other than GbCCHa-2 signaling during the effective period of RNAi after treatment with ds*GbCCHa-2R*, while GbCCHa-2 peptide application elicited an acute response. Neither of these two treatments affected the energy storage in fat body, suggesting that GbCCHa-2 signaling is essential for the lipid and carbohydrate homeostasis in the hemolymph but non-essential for the energy storage processes in the fat body. Low expression levels of *GbCCHa-2* and *-2R* in fat body (Figure 1) may be associated with a lack of correlation between GbCCHa-2 signaling and metabolism occurring in this tissue.

Although ds*GbCCHa-2R* and peptide injection altered the nutritional status by fluctuation of lipids and carbohydrates in the hemolymph, the treatments did not affect the ability to recognize the contents and types of the three major macronutrients in the diets (Figure 6). This result apparently seems to be similar to the fact that the *AKH* mutants of *D. melanogaster* whose hemolymph carbohydrate level is altered [38], do not show shifts of sugar preference [39]. As observed in the current data, the altered levels of hemolymph lipids and carbohydrates after manipulating GbCCHa-2 signaling were relatively smaller than those seen after manipulating AKH signaling in crickets [40]. This is an unusual observation since the compensatory feeding motivation is activated by the general regulatory feeding mechanism to maintain the normal nutritional levels as described previously [36]. There may be a threshold level over the abnormal nutritional status, capable of transducing signals to the CNS to drive the regulatory mechanisms for the suitable and required food choices. However, current data suggest that manipulation of CCHa-2 signaling via *GbCCHa-2R* knockdown and GbCCHa-2 application might not be over the threshold level necessary to induce behavioral changes in the food choice.

In conclusion, GbCCHa-2 signaling suppresses food intake, and is involved in the regulation of metabolism via modulating the concentrations of lipids and carbohydrates in the hemolymph despite little behavioral changes in the food choice. This study provides a new insight into the regulatory network of neuropeptides associated with energy homeostasis in insects through endocrine factors.

## Figures and Tables

**Figure 1 insects-13-00324-f001:**
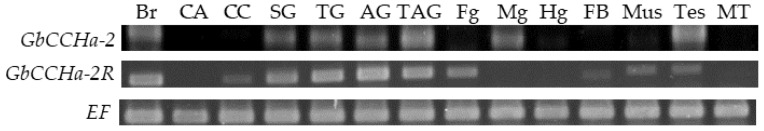
RT-PCR analyses of the tissue distributions of *GbCCHa-2* and -*2R* in male eighth-instar nymphs. *Elongation factor* (*EF*) was used as the internal control gene. No-RT negative control and no-template control yielded no amplification products (Appendix A). Br, brain; CA, corpora allata; CC, corpora cardiaca; SG, subesophageal ganglion; TG, thoracic ganglia; AG, abdominal ganglia; TAG, terminal abdominal ganglion; Fg, foregut; Mg, midgut; Hg, hindgut; FB, fat body; Mus, muscle; Tes, testis; MT, Malpighian tubules.

**Figure 2 insects-13-00324-f002:**
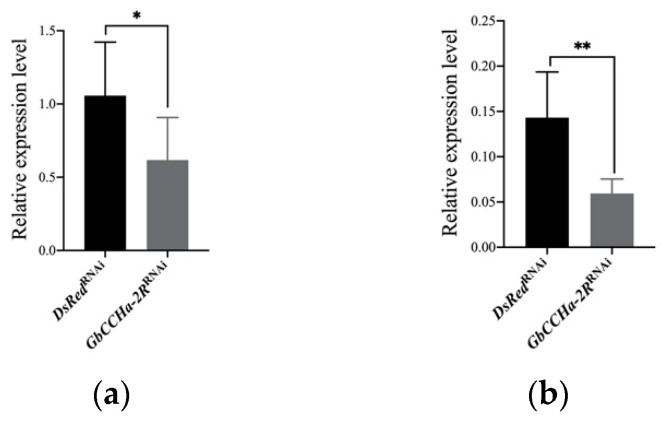
RNAi efficiency. The relative mRNA levels of *GbCCHa-2R* in CNS (**a**) and gut (**b**) on the third day after dsRNA injection. *ß-actin* was used as the reference gene. Values are shown as mean + S.D., n = 5–9, * *p* < 0.05, ** *p* < 0.01, unpaired *t*-test.

**Figure 3 insects-13-00324-f003:**
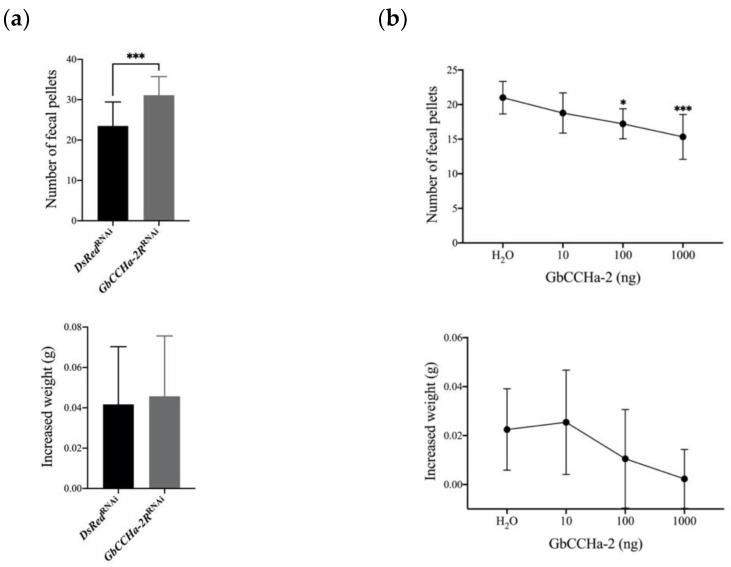
Effects of GbCCHa-2 and ds*GbCCHa-2R* on number of feces and weight gain: (**a**) Effects of *GbCCHa-2R* knockdown on the numbers of fecal pellets representing the amount of food intake for 24 h of the third day after dsRNA injection and 12 h after peptide application; (**b**) Effects of *GbCCHa-2R* knockdown on the weight gain for 24 h of the third day after dsRNA injection and 12 h after peptide application. In RNAi experiments, values are shown as mean + S.D., n = 15–16, *** *p* < 0.001, unpaired *t*-test. In peptide-injection experiments, values are shown as mean ± S.D., * *p* < 0.05, *** *p* < 0.001, n = 9, one-way ANOVA with post hoc Dunnett’s test.

**Figure 4 insects-13-00324-f004:**
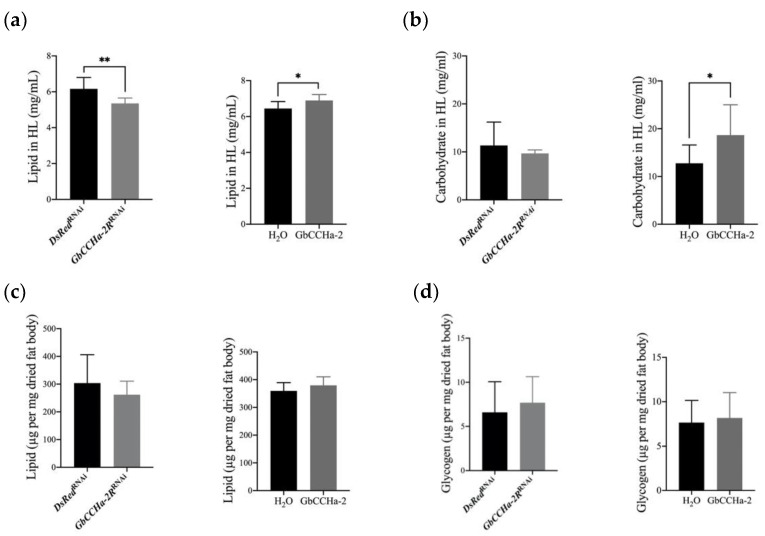
Effects of GbCCHa-2 signaling on the levels of lipids and carbohydrates in the hemolymph and fat body. (**a**,**b**) The levels of lipid (**a**) and free carbohydrates (**b**) in the hemolymph after*GbCCHa-2R* knockdown and GbCCHa-2 administration. (**c**,**d**) The levels of lipid (**c**) and glycogen (**d**) in fat body. Values are showed as mean + S.D., n = 5–9, * *p* < 0.05, ** *p* < 0.01, unpaired *t*-test.

**Figure 5 insects-13-00324-f005:**
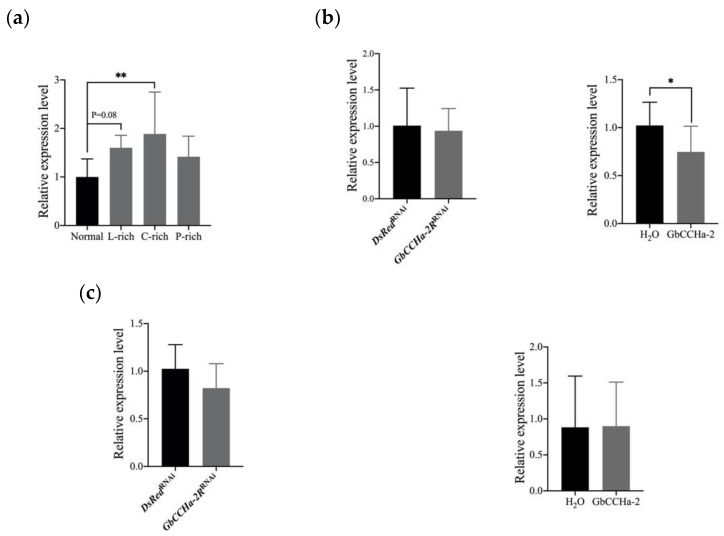
Effects of dietary nutrients on *GbCCHa-2R* and effects of GbCCHa-2 signaling on *Gbilp* and *GbAKH*: (**a**) The transcript levels of *GbCCHa-2R* in the crickets fed on normal diet or lipid-, carbohydrate-, and protein-rich diet (L-, C-, and P-rich); (**b**) The transcript levels of *Gbilp* in CNS after *GbCCHa-2R* knockdown and GbCCHa-2 administration; (**c**) The transcript levels of *GbAKH* in CNS. Values are showed as mean + S.D., n = 6–9, * *p* < 0.05, ** *p* < 0.01, one-way ANOVA with post hoc Dunnett’s test for (**a**), unpaired *t*-test for (**b**,**c**).

**Figure 6 insects-13-00324-f006:**
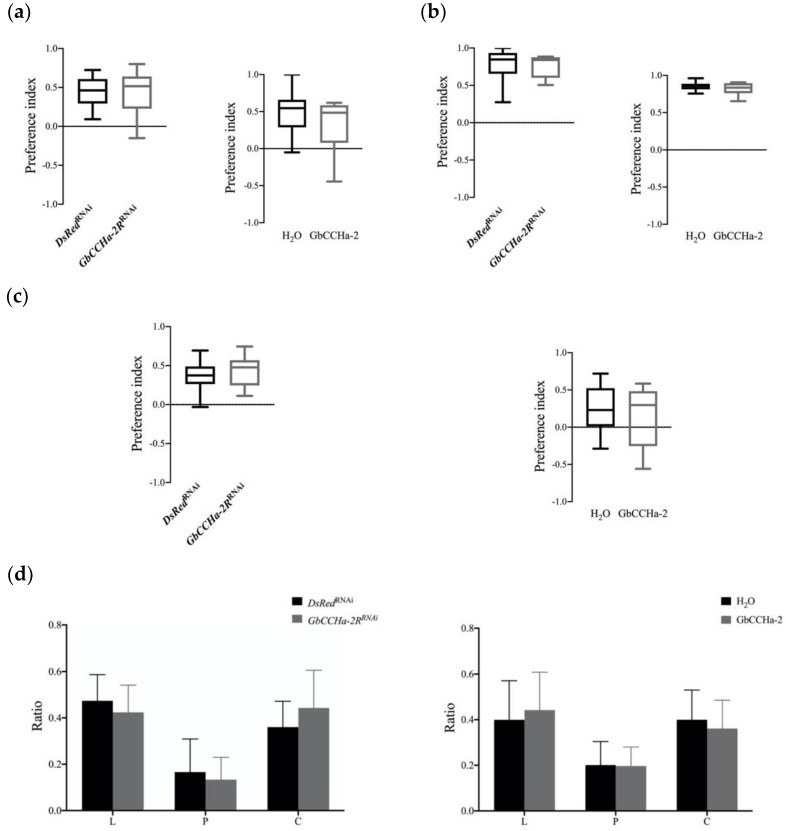
Effects of GbCCHa-2 signaling on food choices: (**a**–**c**) Effects of *GbCCHa-2R* knockdown and peptide injection on the evaluation of the contents of lipids (**a**), carbohydrates (**b**), and proteins (**c**) in diets. Preference index = [W _(rich nutrient)_ − W _(poor nutrient)_]/[W _(rich nutrient)_ + W _(poor nutrient)_]. W, weight of diet eaten; (**d**) Effects of *GbCCHa-2R* knockdown and GbCCHa-2 application on the differentiation of three macronutrient types. Ratio = W _(one nutrient)_/W _(total)_. Values are shown as mean ± S.D., n = 7–11, unpaired *t*-test.

**Table 1 insects-13-00324-t001:** Primers used in this study.

Primer	Sequence 5′-3′
*GbCCHa-2* RT-F	CAGGCAGTAGCAGCAGCA
*GbCCHa-2* RT-R	GAAGCACGAGTGG CCGAA
*GbCCHa-2R* RT-F	CATGGAGGTGGACGTGGAG
*GbCCHa-2R* RT-R	ATGGGGTTGATGCACGAGTT
*EF* RT-F	ATGCCTGTATCTTGACTGCTCA
*EF* RT-R	ATGGTTTGCTTCCAGTTTCAGT
*DsRed2* RT-F	AGAACGTCATCACCGAGTTCAT
*DsRed2* RT-R	CCGATGAACTTCACCTTGTAGA
T7 promoter-F	CTTCTAATACGACTCACTATAG
T7 promoter-R	CTTCTAATACGACTCACTATAG
*GbCCHa-2R* q-F	AGCTGCTCACCTACATCGTG
*GbCCHa-2R* q-R	GAAGATGAACACCAGCGTGC
*Gbilp* q-F	CTGAGAAAGAGCCAGAGCC
*Gbilp* q-R	ATTGCACATGACTTCCGAGA
*GbAKH* q-F	CCCACAGTGCACAGGATCAT
*GbAKH* q-R	CGCCAAAACCAGAACCAAGG
*ß-actin* q-F	TTGACAATGGATCCGGAATGT
*ß-actin* q-R	AAAACTGCCCTGGGTGCAT

## Data Availability

Not applicable.

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
