# Peer review of "CCHamide-2 Signaling Regulates Food Intake and Metabolism in Gryllus bimaculatus"

_insects, 2022, doi:10.3390/insects13040324_

Round 1

Reviewer 1 Report

The manuscript entitled “CCHamide-2 signaling regulates food intake and metabolism in Gryllus bimaculatus” presented interesting results concerning the role of CCHamide in regulation of feeding and lipid and carbohydrate metabolism in insects. It should be highlighted that the high scientific value of the research is related to usage not only of the RNAi method, but also injection of tested peptides. Moreover, the results of these experiments are consistent with each other. Only few questions should be clarified before potential acceptance of the manuscript. Detailed list of comments below:

Lines 101-103. Why Authors did not dissected retrocerebral complex?

Lines 106-108. Always I mentioned the same question. Weather Authors performed no-RT experiment to completely exclude possibility of samples contamination? I know that DNase treatment was performed, but based on my experience, sometimes it is not enough. The results of the no-RT experiment should be added to the supplementary materials.

Line 114. Please add the accession number of mentioned transcriptomic data.

Lines 101-120. Whether the PCR products were sequenced to confirm primers quality?

Lines 147-149. Please add the sequence of synthesized CCHamide and the information about the origin of this knowledge.

Figure 1. Please also present no-template control which should be performed in each PCR experiment.

I know that this is not the fault of Authors, but the quality of Figure 1 does not allow the assessment of the quality of presented PCR products.

Please also specified which was the control for the individuals injected with tested peptide. Whether they were individuals injected with physiological saline or water?

Author Response

We are pleased with the reviewers’ thoughtful comments and suggestions. We have changed almost all indicated points in the revised manuscript. The responses to the comments and suggestions are listed as below.

Lines 101-103. Why Authors did not dissected retrocerebral complex?

>As the reviewer indicated, we have run the RT-PCR using corpora allata and corpora cardiaca as the retrocerebral complex.

Lines 106-108. Always I mentioned the same question. Weather Authors performed no-RT experiment to completely exclude possibility of samples contamination? I know that DNase treatment was performed, but based on my experience, sometimes it is not enough. The results of the no-RT experiment should be added to the supplementary materials.

>As the reviewer indicated, we performed RT-PCR using no-RT sample. Then, we have added the data in the figure.

Line 114. Please add the accession number of mentioned transcriptomic data.

>As the used nucleotide sequence is almost identical with that of TSA sequence in the Genbank. Then, we added the information of the accession numbers for CCHamide-2 and its receptor.

Lines 101-120. Whether the PCR products were sequenced to confirm primers quality?

>As the reviewer implied, we have confirmed the sequence of PCR products. We then added the phrase for this confirmation.

Lines 147-149. Please add the sequence of synthesized CCHamide and the information about the origin of this knowledge.

>As the reviewer indicated, we have added the sequence of the synthesized peptide.

Figure 1. Please also present no-template control which should be performed in each PCR experiment.

>As the reviewer suggested, we added the no-RT control.

I know that this is not the fault of Authors, but the quality of Figure 1 does not allow the assessment of the quality of presented PCR products.

>We renewed the figure.

Please also specified which was the control for the individuals injected with tested peptide. Whether they were individuals injected with physiological saline or water?

>We used water as the control for that experiment.

Reviewer 2 Report

Zhu et al. investigated the function of CCHamide-2 (CCHa-2) signaling in Gryllus bimaculatus. They examined the expression patterns of GbCCHa-2 and GbCCHa-2R in the male nymphs using RT-qPCR. GbCCHa-2 was detected in the central nervous system, midgut, and testes. GbCCHa-2R was widely distributed in the body. To clarify the function of GbCCHa-2 signaling, they knocked down GbCCHa-2R with dsRNA, and injected synthetic CCHa-2 peptide into nymphs. As a result, they found that knockdown of GbCCHa-2R promoted feeding and injection of GbCCHa-2 peptide suppressed feeding in a dose-dependent manner. They then examined metabolic phenotypes of GbCCHa-2R knockdown animals and those injected with GbCCHa-2 peptide. Lipid content in the hemolymph was decreased by GbCCHa-2R knockdown and increased by GbCCHa-2 peptide injection. The amount of carbohydrate in the hemolymph was elevated by GbCCHa-2 peptide injection, while it was not affected by GbCCHa-2R knockdown, probably due to insufficient knockdown. Accumulation of lipids and carbohydrates in the fat body and food preferences were not affected by manipulation of GbCCHa-2 signaling. It has been shown that CCHa-2 signaling has different functions in different insect species, thus the author’s results obtained in Gryllus bimaculatus will provide useful information to researchers in the field of insect endocrinology.

Comments to the authors:

1. It would be interesting to examine the effects of different nutrients to the expression of GbCCHa-2. The results would allow for a better interpretation of the metabolic and feeding phenotypes they observed.

2. To clarify the cause of the metabolic phenotype, insulin synthesis and secretion in GbCCHa-2 knockdown animals or those injected with GbCCHa-2 peptide should be investigated.

3. In the peptide injection experiments, it is better to use a non-specific peptide as a control.

4. The authors need to state how much GbCCHa-2 peptide was used in the metabolic analysis and food preference experiments.

Author Response

We are pleased with the reviewers’ thoughtful comments and suggestions. We have changed almost all indicated points in the revised manuscript. The responses to the comments and suggestions are listed below.

  1. It would be interesting to examine the effects of different nutrients to the expression of GbCCHa-2. The results would allow for a better interpretation of the metabolic and feeding phenotypes they observed.

>We added the experimental data as Figure 5(a). Then, we added the explaining sentences in the “results” session and the caption for the figure. As we mentioned in the previous version of the manuscript, we could not establish the quantitative PCR method for GbCCHa-2.

  1. To clarify the cause of the metabolic phenotype, insulin synthesis and secretion in GbCCHa-2 knockdown animals or those injected with GbCCHa-2 peptide should be investigated.

>We added the experimental data as Figures 5(b) and (c). Then, we added the explaining sentences in the “results” session and the caption for the figure.

  1. In the peptide injection experiments, it is better to use a non-specific peptide as a control.

>As the reviewer indicated, we strongly agreed with the control. However, as the small amount of non-sense peptides sometimes exhibits the different levels of feeding in our laboratory experiences, we always use water as a vehicle and control. Unfortunately, even when we use PBS as the vehicle, the alteration in feeding happens. That is another reason for our usage of water. Then we added the sentence for this indication of usage of water as the control.

  1. The authors need to state how much GbCCHa-2 peptide was used in the metabolic analysis and food preference experiments.

>We used 1 μg of the peptide because of the plateau of the effect of this peptide. We added the sentence for this in the Methods section.

Reviewer 3 Report

CCHamide is a neuropeptide that has been associated with feeding in Drosophila and with post-feeding diuresis in Rhodnius.  This paper studies the role of CCHa-2 in the cricket Gryllus bimaculatus using both RNAi against its receptor CCHa-2R and injection of a synthetic CCHa-2 peptide that they made.  The mRNA for the peptide receptor was found in most tissues, whereas that of the peptide itself was found mainly in the CNS, midgut and testis of the final instar nymph. Although CCHa2-R RNAi treatment significantly increased the number of food pellets eaten, and injection of >100 ng of the peptide decreased intake, no significant weight changes were found. Loss of CCHa-2R function caused a slight decrease in lipid content of the hemolymph but no change in stored lipid nor a significant change in dietary nutrient preference. Injection of the peptide caused increases in both lipid and carbohydrate in the hemolymph but no changes in stored lipid and glycogen in the fat body and no changes in body weight.  They conclude that this peptide in crickets is important in suppressing feeding and homeostasis of hemolymph lipids and carbohydrates. Interestingly, in Drosophila this peptide is found primarily in the gut and its receptor in the brain where the loss of the peptide is associated with reduced feeding in both larvae and adults, the opposite of what is found here for these cricket nymphs.

              The study is well done with good controlled experimental manipulations. The peptide they synthesized has a decent dose-response curve and has effects opposite those seen with the Receptor  RNAi-treated nymphs providing confidence that the synthetic peptide was acting as one would expect.   The data support their conclusions.  It is interesting that this peptide seems to have the opposite role of stimulating feeding in Drosophila and the aphid. Clearly, more work with this peptide in different insects is needed

              Some minor corrections are noted below:

1) The sentence in lines 40-42 seems to have a part missing between ”arthropods” and “by modulating” since the second half of the sentence about invertebrates and vertebrates does not fit with the first half.

2) line 46: do you mean “found” rather than “focused”?

3) line 71-73:  This sentence needs to be rewritten to clarify your meaning.

4) line 84: ….mechanisms underlying feeding…

5) line 198: ….a general rearing diet…

6) Figure 1 in the review copy is not as sharp and crisp as one would like.

7) line 244: I think that you mean “might be dispensable for control of body weight of crickets” since your data says that the peptide does not have any effect on body weight.

8)  The sentence in lines 384-86 is unclear in its meaning.  Please rewrite.

9) lines 400-401:  Do you mean “by adjusting the efficiency of uptake of nutrients from the consumed and ingested food”?  The sentence is unclear.

10) lines 419-20:  I think that you mean the following:  “the altered levels of hemolymph lipids and carbohydrates after manipulating GbCCHa-2 signaling were relatively smaller than those seen after manipulating AKH signaling in crickets.

11) lines 424-25:  Do you mean “….application might not be over the threshold level necessary to induce behavioral changes in the food choice.” ?

Author Response

We are pleased with the reviewers’ thoughtful comments and suggestions. We have changed almost all indicated points in the revised manuscript. The responses to the comments and suggestions are listed below.

1) The sentence in lines 40-42 seems to have a part missing between ”arthropods” and “by modulating” since the second half of the sentence about invertebrates and vertebrates does not fit with the first half.

>As the reviewer indicated, we removed “in arthropods” from the first half sentence.

2) line 46: do you mean “found” rather than “focused”?

>We changed as the reviewer suggested.

3) line 71-73:  This sentence needs to be rewritten to clarify your meaning.

>We rewrote this sentence.

4) line 84: ….mechanisms underlying feeding…

>We changed as the reviewer suggested.

5) line 198: ….a general rearing diet…

>We changed as the reviewer suggested.

6) Figure 1 in the review copy is not as sharp and crisp as one would like.

>We replaced the figure.

7) line 244: I think that you mean “might be dispensable for control of body weight of crickets” since your data says that the peptide does not have any effect on body weight.

>We changed as the reviewer suggested.

8)  The sentence in lines 384-86 is unclear in its meaning.  Please rewrite.

>We rewrote the sentences. We included the discussion content of the additional experiment that the other reviewer suggested.

9) lines 400-401:  Do you mean “by adjusting the efficiency of uptake of nutrients from the consumed and ingested food”?  The sentence is unclear.

>We changed as the reviewer suggested.

10) lines 419-20:  I think that you mean the following:  “the altered levels of hemolymph lipids and carbohydrates after manipulating GbCCHa-2 signaling were relatively smaller than those seen after manipulating AKH signaling in crickets.

>We changed as the reviewer suggested.

11) lines 424-25:  Do you mean “….application might not be over the threshold level necessary to induce behavioral changes in the food choice.” ?

>We changed as the reviewer suggested.

Round 2

Reviewer 1 Report

I glad that Authors responded to all of my comments. The manuscript should be accepted to publication in the current form. 

Reviewer 2 Report

The manuscript has been improved and I believe it is suitable for publication in Insects.